# Selected Atherosclerosis-Related Diseases May Differentially Affect the Relationship between Plasma Advanced Glycation End Products, Receptor sRAGE, and Uric Acid

**DOI:** 10.3390/jcm9051416

**Published:** 2020-05-10

**Authors:** Bogna Gryszczyńska, Magdalena Budzyń, Dorota Formanowicz, Maria Wanic-Kossowska, Piotr Formanowicz, Wacław Majewski, Maria Iskra, Magdalena P. Kasprzak

**Affiliations:** 1Department of General Chemistry, Chair of Chemistry and Clinical Biochemistry, Poznan University of Medical Sciences, Rokietnicka 8, 60-806 Poznan, Poland; magdalena.budzyn@wp.pl (M.B.); iskra@ump.edu.pl (M.I.); magdarut@ump.edu.pl (M.P.K.); 2Department of Clinical Biochemistry and Laboratory Medicine, Poznan University of Medical Sciences, Rokietnicka 8, 60-806 Poznan, Poland; doforman@ump.edu.pl; 3Department of Nephrology, Transplantology and Internal Medicine, Poznan University of Medical Sciences, Przybyszewskiego 49, 60-355 Poznan, Poland; mwanick@ump.edu.pl; 4Institute of Computing Science, Poznan University of Technology, Piotrowo 2, 60-965 Poznan, Poland; Piotr.Formanowicz@cs.put.poznan.pl; 5Institute of Bioorganic Chemistry, Polish Academy of Sciences, Noskowskiego 12/14, 61-704 Poznan, Poland; 6Department of General and Vascular Surgery, Poznan University of Medical Sciences, Dluga 1/2, 61-848 Poznan, Poland; waclaw.majewski@nopb.pl

**Keywords:** oxidatively modified proteins, receptor sRAGE, uric acid, abdominal aortic aneurysms, aortoiliac occlusive disease, chronic kidney disease, atherosclerosis

## Abstract

Our study aimed to identify the relationship between advanced glycation end products (AGEs), soluble receptor for advanced glycation end products (sRAGE), the AGEs/sRAGE, and uric acid (UA) levels in selected atherosclerosis diseases, i.e., abdominal aortic aneurysms (AAA), aortoiliac occlusive disease (AIOD), and chronic kidney disease (CKD), resulting from apparent differences in oxidative stress intensity. Furthermore, we suggest that increased AGEs levels may stimulate an antioxidant defense system reflected by the UA level. The studied group size consisted of 70 AAA patients, 20 AIOD patients, 50 patients in the pre-dialyzed group (PRE), and 35 patients in the hemodialyzed group (HD). The enzyme-linked immunosorbent assay was used to measure AGEs and sRAGE levels. We found a significantly higher concentration of AGEs in CKD patients as compared to AAA and AIOD patients. Furthermore, the sRAGE level was higher in the CKD patients in comparison to AIOD and AAA patients. UA level was significantly higher in the PRE group compared to AAA patients. In conclusion, the diseases included in this study differ in the anti- and prooxidant defense system, which is reflected in the relations between the AGEs, the sRAGE, the AGEs/sRAGE ratio, as well as the UA levels.

## 1. Introduction

Oxidative stress is strongly associated with numerous chronic diseases, such as diabetes mellitus, atherosclerosis, chronic renal failure, neurodegenerative diseases, chronic obstructive pulmonary disease, and cancer [1,2]. It is well known that oxidatively modified proteins are products of reactions between amino acid side chains and reactive oxygen species (ROS) or reactive nitrogen species (RNS), respectively. Furthermore, non-oxidative modification of proteins, known as glycation, also occurs. The reaction of reducing carbohydrates with amine groups of proteins leading to the formation of AGEs.

The growing number of studies confirm that AGEs may be involved in the pathogenesis of various diseases, particularly those in which atherosclerosis constitutes the main contributing factor. Their mechanisms of action are not well understood, although they probably result from their interaction with the receptor for advanced glycation end products (RAGE) [3,4,5]. RAGE is found on the surface of numerous cells, including monocytes, macrophages, endothelial cells, smooth muscle cells, and fibroblasts [2]. Moreover, it has been confirmed that some of the intracellular signal transduction pathways are activated by the RAGE receptor [3,6]. The complex formation between RAGE and AGEs activates an inflammatory cascade that stimulates gene transcription for cytokines and growth factors (TNF-α, IL-1), as well as for adhesion molecules (ICAM-1, VCAM-1) [2,7]. In contrast, the soluble isoforms of RAGE (sRAGE) present in the circulation are capable of limiting the signaling pathway activated by the membrane-standing RAGE receptor [2,8,9]. The sRAGE binds with the AGEs and prevents their interaction with receptors. Since sRAGE can bind with AGEs and neutralize their negative effects in the simultaneous measurement of AGEs and sRAGE, it seems to be more appropriate for establishing the possible function of the AGE-RAGE axis in various diseases [4,10,11]. Some studies indicated that a critical balance between AGEs and sRAGE might be impaired, resulting in the promotion of negative effects caused by AGEs, such as inflammation and enhanced oxidative stress. Although the exact mechanism of the interaction between AGEs and sRAGE is well understood, the number of diseases in which the critical balance between these two factors has been evaluated is still limited. 

In the past decade, numerous epidemiological studies were focused on the cardiovascular effects of UA and its role in cardiovascular diseases. It has been suggested that a high serum level UA is beneficial due to the antioxidant properties against pro-oxidant molecules [12,13]. On the other hand, some researchers have demonstrated relationships between UA metabolism and the progression of various disorders, such as hypertension, diabetes mellitus, CKD, and heart failure [13,14,15]. Interestingly, there is a large body of evidence that high UA levels stimulate the RAGE-ligand axis, which plays an essential role in the pathogenesis of atherosclerosis [15,16]. There is hardly any literature data on sRAGE and AGEs concerning hyperuricemia closely associated with renal oxidative stress in CKD, as well as in AAA and AIOD patients who usually present normal renal function. Hence, it is interesting to determine whether the AGEs level can stimulate UA synthesis in order to help maintain the AGEs-sRAGE balance. 

The purpose of our study was to test the hypothesis that AAA, AIOD, and CKD affect the relationship between AGEs, sRAGE, as well as UA levels due to apparent differences in oxidative stress intensity. Furthermore, we suppose that the increased levels of AGEs may stimulate not only the expression of the sRAGE scavenger receptor but also an antioxidant defense system in response to increased oxidative stress reflected by UA level. 

Therefore, we evaluated AGEs and sRAGE levels, as well as AGEs/sRAGE ratio in different groups of patients, including those with cardiovascular diseases (AAA and AIOD patients), as well as those with CKD, including pre-dialysis patient group (PRE) and hemodialysis patient group (HD). The second objective was to estimate how these parameters were affected by patients’ age, sex, inflammatory state, and kidney function. Additionally, the relation between AGEs and sRAGE, as well as the AGEs/sRAGE ratio and the serum UA level was analyzed.

## 2. Material and Methods

The present study is a continuation of previous research in which the approval number of the Bioethical Commission, the research criteria, exclusion criteria, and clinical examination were described [17]. The essential details on the studied groups are listed below.

### 2.1. Patients

#### 2.1.1. AAA and AIOD Patients

The size of AAA and AIOD group was increased as compared to the previous study and was as follows: 70 patients in the AAA group (55 men and 15 women; mean age 70.25 ± 8.65) and 20 patients in the AIOD group (14 men and six women; mean age 63.78 ± 6.80), respectively. Patients were admitted to the Department of General and Vascular Surgery at Poznan University of Medical Sciences, Poznan, Poland. The Doppler ultrasonography, computed tomography, or arteriography were performed in all AAA and AIOD patients [17]. The mean internal diameter of the aneurysms in the AAA patients was 61.66 ± 12.11 mm.

#### 2.1.2. CKD Patients

The study was carried out in a group of 85 CKD patients treated at the Department of Nephrology, Transplantology, and Internal Medicine at the Poznan University of Medical Sciences. The severity of CKD was identified on the basis of the measurement of the estimated glomerular filtration rate (eGFR). Depending on the severity, patients were assigned to a pre-dialysis patient group (PRE), or a hemodialysis patient group (HD). The PRE group (stage 3–4) included patients (*n* = 50) with severely reduced kidney function (eGFR 24.53 ± 9.35 mL/min/1.73 m^2^). In contrast, the HD group (stage 5) consisted of 35 patients with significant, severely reduced kidney function, who underwent maintenance HD for at least 12 months (eGFR 7.29 ± 3.17 mL/min/1.73 m^2^) [17]. The clinical data with highlighted cardiovascular risk factors and the diagnosed concomitant diseases, as well as the medications administered to the patients, are listed in Table 1. In contrast, Table 2 presents the biochemical characteristics of all patients included in the study.

### 2.2. Sample Collection

Blood samples were collected from the arms of AAA, AIOD, PRE, and HD patients in the recumbent position after 10 min of rest into heparin anticoagulant tubes. The preparation, as well as plasma samples storage conditions, were described in the previous study [17]. 

### 2.3. Laboratory Analysis

#### 2.3.1. AGEs Assay Kit (Cell Biolabs, Inc., San Diego, CA, USA)

The first step for the AGEs measurement was to coat the plate by an AGE conjugate coating solution. Next, advanced glycation end products-bovine serum albumin (AGE-BSA) standards and analyzed samples were applied on the pre-coated ELISA plate. After incubation, an anti-AGE polyclonal antibody and a Horseradish Peroxidase (HRP)-conjugated secondary antibody were added. Finally, absorbance was measured at 450 nm using Zenyth 200 Microplate Spectrophotometer (Anthos Labtec Instruments GmbH, Salzburg, Austria). The concentration of AGE adducts was calculated based on the AGE-BSA standard curve. The concentration of AGEs was expressed in (μg/mL). 

#### 2.3.2. Receptor sRAGE (RayBiotech, Norcross, Peachtree Corners, GA, USA)

Initially, standard solutions and samples were prepared according to the manual and then added into appropriate wells coated by the human RAGE antibody. Next, the plate was incubated and washed out. Next, the biotinylated antihuman RAGE antibody was pipetted into each well. Finally, a Horseradish Peroxidase (HRP)-conjugated streptavidin, followed by the 3,3,5,5′-tetramethylbenzidine (TMB), was added into each well. The absorbance was measured at 450 nm using Zenyth 200 Microplate Spectrophotometer (Anthos Labtec Instruments GmbH). The serum sRAGE level was expressed in (pg/mL).

#### 2.3.3. hsCRP (DRG International Inc., Springfield Township, NJ, USA)

A solid-phase enzyme-linked immunosorbent assay determined the hsCRP concentration. Initially, unknown samples and a control sample were diluted 100-fold according to the manual. Subsequently, standards, samples, and control samples followed by the C-reactive protein (CRP) Enzyme Conjugate Reagent were dispensed into each well. Next, the plate was incubated at room temperature for 45-min, followed by washing the wells. Then, 3,3,5,5′-tetramethylbenzidine (TMB) was added, and the reaction was stopped by adding 1 N HCl. The absorbance was measured at 450 nm using Zenyth 200 Microplate Spectrophotometer. The hsCRP level was calculated based on the CRP standard curve.

### 2.4. Statistical Analysis

Statistical analyses were performed using GraphPad InStat or the GraphPad Prism software 8.0 (Graph-Pad Software, San Diego, CA, USA). The following statistical tests were used to verify the normal distribution of data—the Kolmogorov–Smirnow, Shapiro–Wilk, and Pearson omnibus normality test. However, the Pearson omnibus normality test was given the highest priority according to the producer recommendation. The comparison of four studied groups was performed using one-way ANOVA analysis. In this case, either the Kruskal–Wallis test followed by Dunn’s Multiple Comparison Test, or one-way analysis of variance was used, depending on the data distribution. In contrast, the Student’s test, or Mann–Whitney test were used for the comparison of two analyzed groups depending on the data distribution. The strength of the associations between different variables was tested using either the Pearson or Spearman correlation coefficients. Additionally, in order to provide greater statistical power and to estimate the independent influence of selected factors on analyzed parameters, multiple linear regression was carried out in the studied groups. In all cases, *p*-value ≤ 0.05 was considered statistically significant.

## 3. Results

### 3.1. AGEs, sRAGE, AGEs/sRAGE Ratio, and UA Level in the Studied Groups

Significant differences were observed for AGEs, sRAGE, AGEs/sRAGE, and UA levels between the studied groups. 

The data presented in Figure 1 show significantly higher concentrations of AGEs in the HD group (median: 2978 μg/mL; range: 1603–4939; mean ± SD: 3015 ± 1772 μg/mL) as compared to the AAA (median: 18.00 μg/mL; range: 10.40–35.20; mean ± SD: 34.65 ± 41.99 μg/mL) and AIOD (median: 27.58 μg/mL; range: 17.00–49.07; mean ± SD: 44.00 ± 44.75 μg/mL) patients. An identical observation was made in the PRE group; an elevated concentration of AGEs (median: 1638 μg/mL; range: 1158–2103; mean ± SD: 1864 ± 1216 μg/mL) was seen in comparison to AAA and AIOD patients. However, insignificant differences in the AGEs level between AAA and AIOD, as well as PRE and HD, were observed. The values of AGEs in the studied groups may be listed in descending order: HD > PRE > AIOD > AAA. 

The median values of sRAGE were as follows—77.0 pg/mL (range: 35.7–133.5) in AIOD patients; 30.7 pg/mL (range: 30.7–107.6) in the AAA group; 3117 pg/mL (range: 2214–4016) in PRE patients and 4004 pg/mL (range: 2586–4097) in the HD group, respectively. The level of sRAGE was higher in the HD group in comparison to AIOD and AAA. The values of sRAGE in the studied groups may be listed in the following descending order: HD ≈ PRE > AIOD ≈ AAA (Figure 2). 

Finally, the ratio of AGEs/sRAGE was calculated (Figure 3). A significantly lower AGEs/sRAGE ratio was found for the AIOD group (median: 0.34 μg/pg; range: 0.14–0.85) and AAA patients (median: 0.40 μg/pg; range: 0.18–0.60), compared to the HD group (median: 0.87 μg/pg; range: 0.60–1.23). A trend of an increasing AGEs/sRAGE ratio was observed, starting with AIOD patients, followed by AAA patients, and ending with the PRE (median: 0.56 μg/pg; range: 0.30–0.80) and HD group.

The data presented in Figure 4 show the significantly higher UA level in the PRE group (386.6; range: 316.7–450.6) compared to the AAA patients (352.0; range 302.5–384.5). No difference in UA concentration was reported between patients in the PRE and HD (374.7; range: 321.2–404.5) groups, as well as between the PRE and AIOD patients (351.0; range: 272.3–370.3). 

### 3.2. The association of AGEs, sRAGE, AGEs/sRAGE Ratio, and UA with Age, Gender, hsCRP, as well as estimated glomerular filtration rate (eGFR) in Studied Groups

In the present study, we determined the influence of gender on AGEs, sRAGE, the AGEs/sRAGE ratio, and the levels of UA in the studied groups (Table 3). In both male and female patients in each of the studied groups, all the analyzed parameters did not differ significantly apart from AGEs/sRAGE in AAA.

Subsequently, in order to assess the relationship between the analyzed parameters as well as their association with age, hsCRP, and eGFR, a univariate analysis was performed in each group separately. Then, it was run independently in the CKD group, including the PRE and HD patients, as well as in the CVD group, including the AAA and AIOD patients. In the AAA group, a significant correlation was found for AGEs and sRAGE, whereas a negative relationship was demonstrated for sRAGE and eGFR. In the PRE group, the AGEs concentration and sRAGE levels were found positively correlated with eGFR. Furthermore, in the HD group, the AGEs level was found to be positively correlated with the sRAGE and the AGEs/sRAGE ratio but negatively correlated with the UA level. Univariate analysis revealed that AGEs correlated positively with sRAGE in CKD patients. In the CVD group, eGFR negatively correlated with the sRAGE and UA levels. All statistically significant correlation coefficients, *p* values, and 95% confidence intervals in the studied groups are presented in Table 4. Insignificant correlation coefficients have not been included. 

Moreover, in order to provide higher statistical power and to demonstrate the independent influence of age, gender, hsCRP, and eGFR on AGEs, sRAGE, and UA, multiple linear regression was conducted in the CKD group (PRE and HD), the CVD group (AIOD and AAA), and among the whole group of patients (Table 5). In the CKD group, sRAGE was independently associated with AGEs, whereas UA, as well as AGEs with eGFR. In the CVD group, independent of the other factors considered, the UA level was associated with eGFR, whereas sRAGE with AGEs and eGFR. In a multivariate analysis, taking into consideration all the studied groups, AGEs and eGFR were associated with an increased sRAGE concentration independent of age, gender, and hsCRP. Furthermore, AGEs and UA levels were independently associated with eGFR. 

### 3.3. The Association of AGEs and UA Level with the Diameter of the Aneurysm in AAA Patients

Additionally, in the present study, the median value of the aneurysm diameter was used as the criterion allowing for the division of AAA patients into appropriate groups with a low (<62 mm) and a high (≥62 mm) diameter. The classification of AAA patients into relevant subgroups, according to the aneurysm diameter, did not reveal a significant difference in AGEs, sRAGE, AGEs/sRAGE ratio, and UA level between patients with low diameter vs. patients with high diameter. However, a positive correlation was found for the AGEs level and aneurysm size in the AAA group with a low (<62 mm) diameter (Figure 5). In addition, it was also demonstrated that the UA concentration correlated negatively with the aneurysm size in AAA patients with a high (≥62 mm) diameter (Figure 6). 

## 4. Discussion

Atherosclerosis constitutes a chronic disease process involving numerous cellular factors, such as endothelial cells, myocytes, lymphocytes, and platelets, as well as cytokines and adhesion molecules [18]. In the last decades of research, at least two types of atherosclerosis were identified—the “classic” (typical for AIOD and AAA), and the “non-classic” (typical for PRE and HD) [19,20]. These types differ not only in the mechanisms underlying the formation of plaques but also in the intensity of the associated oxidative stress [17]. It should be stressed that oxidative stress, plays a more significant role in CKD, particularly in HD patients than in CVD [20]. It is due to insufficient renal function resulting in increased circulating uremia-specific toxins and AGEs accumulation. Moreover, cyclical contact of the blood with the dialysis membrane may lead to the enhanced production of free radicals [21]. 

In the present study, AGEs levels in plasma were found to be significantly lower among both AIOD and AAA patients compared to the PRE and HD patients. The significant difference in the AGEs concentration in the “arterial” patients’ vs. CKD patients who are known to be burdened by higher production of reactive oxygen species (ROS) may indicate that AGEs level rises according to the intensity of the oxidative stress. Moreover, the lower value of AGEs observed in AIOD and AAA may be due to the normal renal function providing efficient AGEs excretion and preventing their accumulation. However, it must be underlined that in both CVD as well as CKD, AGEs contribute to atherosclerosis development [22,23]. The pathophysiological mechanism described by Vlassara et al., as well as Oleniuc et al., involves a progressive rigidity of proteins due to cross-linking of proteins, cytokine, and increased adhesion molecules expression [24,25]. Consequently, the identification of AGEs in the atherosclerotic plaques of patients with CKD confirms the link between AGEs and atherosclerosis [26]. 

The damaging effect of AGEs is the result of their chemical and biological features, as well as their interaction with particular receptors, such as RAGE [3,5]. The soluble isoform of RAGE, known as sRAGE, plays various functions, including limiting the side effects caused by the interaction of AGEs with their receptors. In the present study, the level of sRAGE was up to 10-times higher in both the CKD subgroups as compared to AAA and AIOD patients. It was also demonstrated that the sRAGE values in the PRE group differed insignificantly when compared to the HD patients. Interestingly, the obtained findings indicate that the level of sRAGE in CKD-5 patients was significantly higher than in the CKD 3–4 individuals, which may be a mechanism of protection against increased oxidative stress as well as inflammation [27]. On the contrary, there was a lack of difference in the sRAGE levels between the PRE and HD groups in the present study. Two possible mechanisms may explain this phenomenon. The first one assumes an increased sRAGE clearance in the HD patients due to the repeating cyclical hemodialysis; the second one describes the removal of sRAGE due to albuminuria accompanying acute renal failure. Both processes explain a decreasing concentration of sRAGE in HD patients, also demonstrated in our study by the lack of difference in its levels between the HD and PRE patients. In our study, we also demonstrated a significantly lower sRAGE level in AAA and AIOD patients compared to CKD patients, which may indicate weaker oxidative stress and inflammation in CVD, resulting in weaker antioxidant and anti-inflammatory responses in this group of diseases. 

The AGEs/sRAGE ratio best illustrates the relationship between AGEs and sRAGE. It is not only classified as an independent biomarker for all AGE-RAGE–associated diseases, including diabetes and renal disease but also as a parameter reflecting the function of the AGE-RAGE axis in various diseases [28,29]. In the present study, a significantly higher AGEs/sRAGE ratio in the HD patients rather than in the AAA and AIOD groups was found. Furthermore, a significantly lower AGEs/sRAGE ratio in the AIOD group compared to PRE patients was demonstrated. Thus, our results may suggest that the concentration of sRAGE in CKD patients is inadequate. Furthermore, the higher level of AGEs per sRAGE may reflect the interaction of ligands with the RAGE receptor, which is more facilitated in CKD patients in comparison to AAA and AIOD patients. 

In the previous studies, the prooxidant-antioxidant status in the AIOD patients was insufficiently investigated. Although some researchers indicated that ROS levels significantly increased with AIOD severity, no literature data has analyzed AGEs levels in the blood of the AIOD patients [30]. The same applies to AAA patients. However, there are some studies, including AAA patients, which revealed that pentosidine concentrations, as well as cross-linking AGEs in-wall biopsies, were increased in diabetic compared with non-diabetic AAA individuals. However, there was no simultaneous analysis of AGEs in AAA patients’ blood [31]. In the present study, for the first time, we have demonstrated that AAA and AIOD promote the generation of AGEs compared to CKD in a different way, i.e., due to the type of atherosclerosis, as well as the oxidative stress intensity and localization [32,33]. Based on our study, it is possible to assume that the glycation process is more intense in the PRE and HD patients due to the additional complexities involved in CKD pathogenesis. Furthermore, the progressive deterioration of renal function with increased severity of CKD needs to be particularly highlighted. The AIOD and AAA patients, to a large extent, are characterized by proper kidney function reflected in lower values of renal function parameters listed in Table 1 and Table 2. Finally, in the process of designing the study, diabetes mellitus patients were excluded, since an elevated glucose level may induce protein glycation. 

In the present study, the UA level in plasma was found to be significantly lower among the AAA patients in comparison to the PRE patients. Hyperuricemia is typical in CKD; however, data concerning the long-term outcomes of the CKD patients, as well as the pro- or antioxidant function of UA found in the literature are inconclusive [34,35]. Several studies demonstrated the role of UA as an independent CVD and CKD risk factor [36,37]. Moreover, a high UA level was associated with lower mortality in patients with CKD-5 receiving hemodialysis compared to the CKD non-dialysis patients [38]. It was suggested that an elevated UA concentration might improve cardiac protection and better survival rates of HD patients [38]. Interestingly, PRE and HD differed insignificantly in the serum UA level (Figure 4). In our opinion, the cyclical hemodialysis improves the clearance of small water-soluble non-protein-bound molecules like UA [39]. The significantly lower UA level in the AAA patients compared to the PRE group indicates sufficient renal function, as demonstrated by data presented in Table 1. 

We also found a positive correlation between the AGEs level and sRAGE concentration in the AAA and HD patients, as well as in the CKD group. Our results are consistent with the observations made by El-Saeed et al., who demonstrated a positive correlation between the parameters mentioned above in children undergoing hemodialysis [40]. In general, the positive interrelationship between AGEs levels rather than the inverse relationship with sRAGE in humans, was observed [40,41]. On the basis of the relationship mentioned above, we may conclude that AGEs stimulate the generation of sRAGE. The negative effect of the renal dysfunction on the antioxidant defense system is confirmed by the negative correlation between eGFR and sRAGE in the PRE group. The proper renal function, reflected by a higher eGFR value, is strongly associated with an increased value of sRAGE, probably due to the fact that sRAGE excretion is not impaired. The multivariate linear analysis confirmed that age, gender, and hsCRP are not the factors that influence AGEs, sRAGE, as well as UA levels in all the studied groups. Furthermore, the multivariate linear regression analysis revealed that independently of other factors such as age, gender, inflammatory state, eGFR is associated with AGEs level in all the studied patients, as well as in the CKD patients. Moreover, the independent relationship between eGFR and AGEs confirms a significant role of impaired kidney function in the enhancement of oxidative stress in CKD patients. In addition, the increase in AGEs level and a corresponding rise in sRAGE concentration indicate intensified generation of scavenging receptor for the protection against rising oxidative stress in CKD, CVD, as well as in all the patients taken together. The high AGEs level independently stimulates the liberation of the sRAGE receptor, which may represent the adaptive antioxidant mechanism against further oxidation of biomolecules.

The dual function of UA as an antioxidant and as a prooxidant molecule has been widely discussed in the literature [13,15,42]. UA can neutralize prooxidants, such as hydrogen peroxide, peroxynitrite and hydroxyl radical; it also shows a scavenging potential against O_2_^−^ in the plasma [15,43,44]. However, despite the beneficial effects of UA, many researchers have pointed the relationship between the elevated serum UA level and numerous diseases, such as dyslipidemia, atherosclerosis, hypertension, renal failure, diabetes, coronary heart disease [13,14,15,42]. A growing body of evidence indicates that UA induces endothelial dysfunction, stimulates the RAGE signaling pathway and activates nuclear factor kappa B (NF-ĸB), as well as decreases the production of NO, and promotes oxidative stress following the entry to the cell environment [16,45]. We assumed that the increased AGEs level might stimulate not only sRAGE expression but also the UA synthesis. Nevertheless, the multivariate linear analysis did not confirm our hypothesis. In turn, we demonstrated that another factor influences an increased UA level, i.e., renal dysfunction reflected by low eGFR value, which seems to be strongly associated with UA disturbance in the CKD and CVD patients, as well as in all subjects taken together. However, the stimulation of the synthesis of other endogenous antioxidants, not only UA, by an increased AGEs level has not been excluded. Therefore, the measurement of total antioxidant capacity in the studied patients should be evaluated in further studies. Our unconfirmed hypothesis described above does not eliminate the antioxidant potential of UA, as UA represents up to 60% serum-free radical scavenging capacity [46]. On the other hand, it should be emphasized that the UA might increase oxidative stress by means of renal and systemic inflammation [46]. In fact, Cai et al. suggested the mechanism of UA-activated inflammation in endothelial cells [16]. The authors claim that a high UA level enhances the generation and the release of group box chromosomal protein 1 (HMGB1), an inflammatory molecule, which may increase oxidative stress and activate the RAGE signaling pathway resulting in endothelium dysfunction [15,16]. 

A novel finding of our study is the observation that the serum UA level is decreased in AAA patients with aneurysm diameters larger than 62 mm. According to our observations, the negative correlation between the UA level and the aneurysm diameter suggests that patients with a larger aneurysm diameter are protected to a smaller degree against oxidative stress. This, in turn, indicates that in the AAA pathogenesis, the antioxidant activity of UA, i.e., the main serum-free radical scavenger, is overcome by the increased generation of free radicals and ROS. Interestingly, Esen et al. demonstrated a positive correlation between serum UA levels and the ascending aortic diameter in patients with ascending aortic aneurysms (AscAAs) [46]. They found that an increased UA level may be responsible for the elevated serum antioxidant capacity among individuals with AscAA [46]. This inverse relationship found by Esen et al. may imply that the studied group was more protected against oxidative stress due to a different course of the disease or a smaller aneurysm diameter, or both of them. On the other hand, the study on UA in aortic aneurysm specimens revealed a significant increase in UA levels in the wall altered due to atherosclerosis in comparison to a non-atherosclerotic artery [47]. Patetsios and co-authors concluded that UA deposition due to an increased xanthine oxidase activity within the artery wall probably results in damage [47]. 

Interestingly, the positive correlation between AGEs level and aneurysm diameter lower than 62 mm in AAA patients was found. There are no studies regarding the association between AGEs serum level and the AAA severity, reflected by the aneurysm diameter size. We may hypothesize that oxidative stress, which plays an essential role in the pathogenesis of AAA, is probably more intensified in the early stage of the disease development. This may suggest that oxidative stress has a more significant effect on the development/growth of the aneurysm in the early stages of the disease. In fact, the significance of oxidative stress is possibly decreasing in the advanced stage of the disease. It is possible that an adaptive mechanism against further oxidation may have taken place in AAA patients with a larger aneurysm diameter; however, this theory would require additional experiments. It should be emphasized that AGEs significantly contribute to wall stress. Some authors have suggested that AGEs are involved in arterial stiffening and rigidity in the vessel wall [28,29]. It is difficult to determine what would be the AGEs concentration level in the aneurysm walls. However, it can be assumed that AGEs circulating in the blood do not reflect it, due to the influence of kidneys and the liver role in the exposure to and accumulation of AGEs [48]. Furthermore, the AGEs-RAGE imbalance increases endothelial permeability leading to vascular dysfunction. Although we did not analyze a correlation of markers of endothelium damage with AGEs level in present work, however on the basis of our previous studies, we may hypothesize that AGEs play a destructive role in the endothelial layer [4].

## 5. Conclusions

In conclusion, the present study demonstrated that the AGEs, the sRAGE, and the UA levels are independently associated with the renal function reflected by eGFR. Furthermore, the positive correlation between AGEs and sRAGE indicates an important role of AGEs as a factor stimulating sRAGE expression. Consequently, a lack of gradation in AGEs concentration according to the disease severity may suggest more intensified oxidative stress in PRE patients, despite a smaller incidence of complications than in HD patients. Moreover, a higher level of AGEs per sRAGE in CKD patients may result in the inadequate competitive function of the sRAGE against the AGEs. The increased UA level in CVD and CKD patients is rather associated with an impaired kidney function than with a stimulating effect of high AGEs level. The atherosclerosis-related diseases selected for the current study appear to differ in the anti- and prooxidant defense system, reflected in the relations between the AGEs, the sRAGE, the AGEs/sRAGE ratio and the UA levels.

## 6. Limitations of the Study

There are limitations to our study. Firstly, the study was performed in small groups, especially AIOD, and with the unequal number of men and women, which is especially notable in AAA group. This is due to the fact that the male gender constitutes AAA risk factor among the Polish population. Furthermore, the effect of gender on the analyzed parameters was not found in all studied groups. Therefore, despite the small size of the groups and the unequal gender structure, their biochemical and clinical characteristics seem to be well suited to the specificity of these diseases. Another limitation was the potential effect of anti-hypertensive drugs and non-steroidal anti-inflammatory drugs on the level of the studied parameters. Despite ambiguous data, the impact of medications taken by patients on AGEs, sRAGE and UA levels could not be excluded. The hypothesis that an increased AGEs level may stimulate the UA synthesis to help maintain an AGEs-sRAGE balance was not confirmed. It cannot be entirely ruled out that AGEs may stimulate the synthesis of another antioxidant, although additional research is required to verify this concept. 

## Figures and Tables

**Figure 1 jcm-09-01416-f001:**
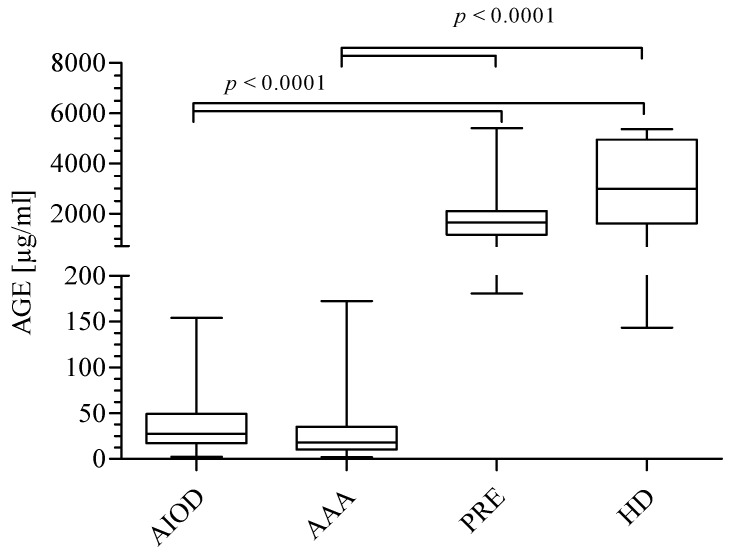
Concentrations of advanced glycation end products (AGEs) in the studied groups expressed as μg/mL. The comparison of the four studied groups was performed using the Kruskal–Wallis test followed by Dunn’s multiple comparison test.

**Figure 2 jcm-09-01416-f002:**
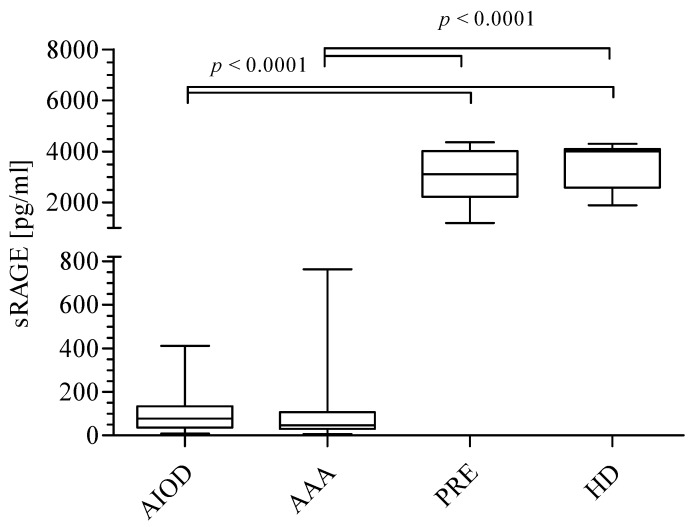
sRAGE level in the studied groups expressed as pg/mL. Data were analyzed using the Kruskal–Wallis test, followed by Dunn’s multiple comparison test.

**Figure 3 jcm-09-01416-f003:**
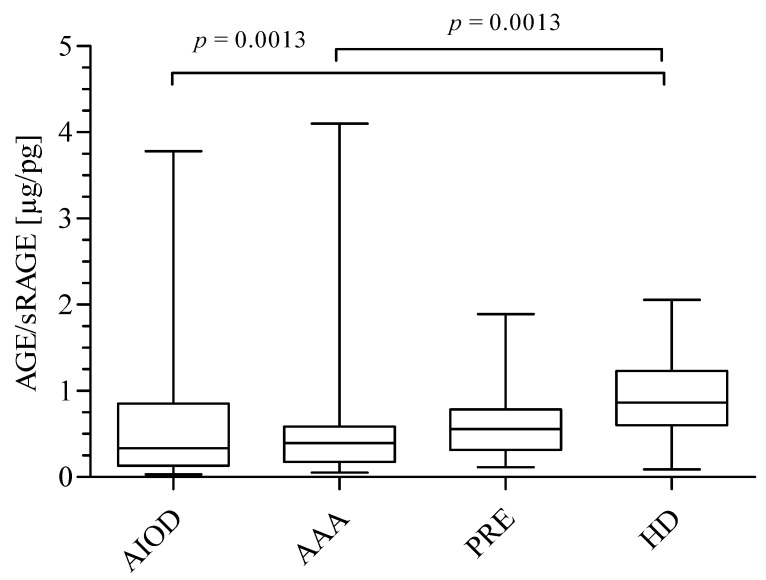
AGEs/sRAGE ratio in studied groups expressed as μg/pg. The comparison of four studied groups was performed using the Kruskal–Wallis test followed by Dunn’s multiple comparison test.

**Figure 4 jcm-09-01416-f004:**
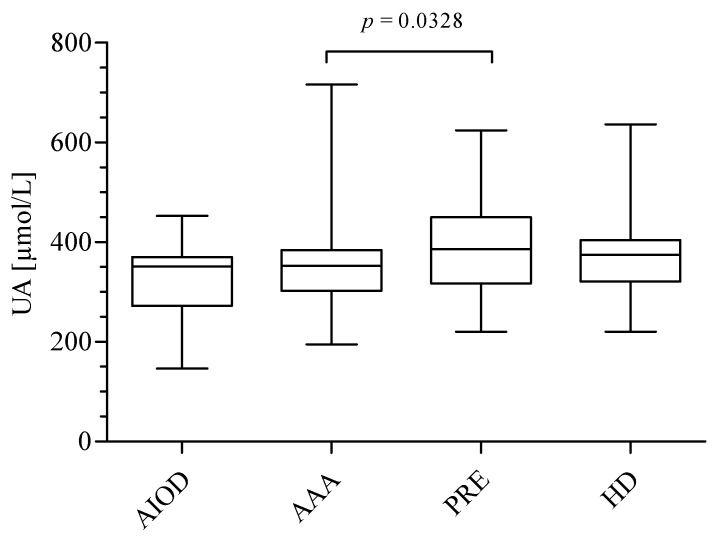
Uric acid (UA) concentration in the studied groups expressed as μmol/L. The Kruskal–Wallis test, followed by Dunn’s Multiple Comparison Test, were used to compare the studied groups.

**Figure 5 jcm-09-01416-f005:**
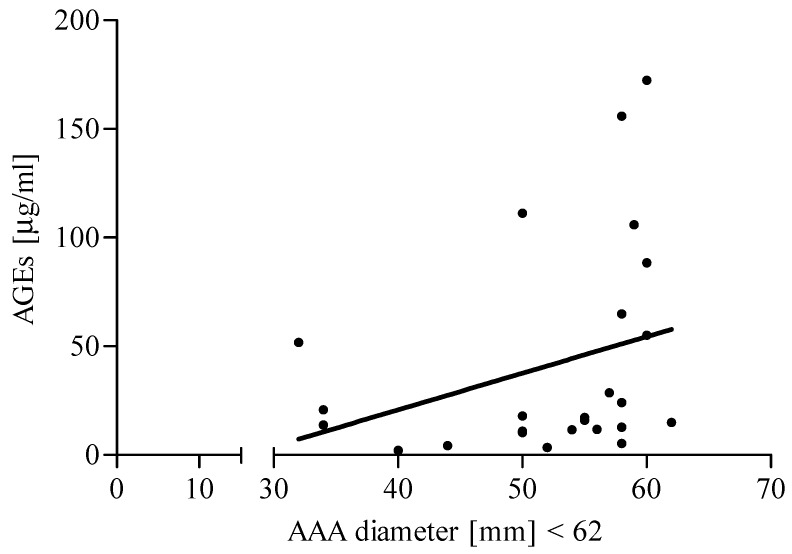
Correlation between AGEs level and aneurysm diameter in abdominal aortic aneurysms (AAA) patients (diameter < 62 mm). Spearman correlation coefficient *r* = 0.4315, *p* = 0.0313.

**Figure 6 jcm-09-01416-f006:**
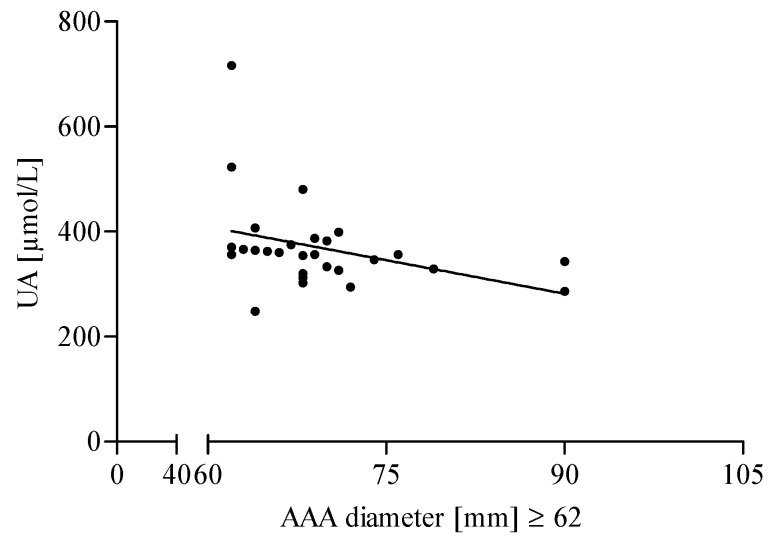
Correlation between uric acid (UA) concentration and aneurysm diameter in AAA patients (diameter ≥ 62 mm). Spearman correlation coefficient *r* = −0.4602, *p* = 0.0137.

**Table 1 jcm-09-01416-t001:** Clinical characteristics of the analyzed groups of patients with AAA, AIOD, PRE, and HD.

Parameters	AAA (70 Patients) No. (%)	AIOD (20 Patients) No. (%)	PRE (50 Patients) No. (%)	HD (35 Patients) No. (%)
Age (mean ± SD)	70.25 ± 8.65	63.78 ± 6.80	71.6 ± 13.12	54.03 ± 16.18
Gender (male/female)	55/ 15	14 / 6	27/23	24/11
Hypertension	48 (68)	12 (60)	50 (100)	34 (100)
Hypercholesterolemia	18 (25)	3 (15)	50 (100)	34 (100)
Coronary artery disease	30 (44)	8 (40)	17 (34)	34 (100)
Previous myocardial infarction	12 (17)	3 (15)	8 (16)	9 (26.5)
Cerebrovascular accident	7 (10)	1 (5)	1 (2)	4 (11.8)
Kidney disease	8 (11.5)	2 (10)	50 (100)	34 (100)
Pulmonary disease	7 (10)	4 (20)	0	0
Medications				
β-blocker	32 (46)	10 (50)	29 (58)	20 (58.8)
ACEIs	35 (50)	8 (40)	25 (50)	11 (32)
Statins	37 (52.8)	15 (70)	39 (78)	27 (79.4)
NSAIDs	70 (100)	20 (100)	40 (80)	11 (32.4)

AAA, abdominal aortic aneurysms; AIOD, aortoiliac occlusive disease; PRE, pre-dialyzed group; HD, hemodialyzed group; ACEIs, angiotensin converting enzyme inhibitors; NSAIDs, nonsteroidal anti-inflammatory drugs.

**Table 2 jcm-09-01416-t002:** Biochemical baseline characteristics of AAA, AIOD, PRE, and HD groups.

Parameters	AAA (70 Patients)	AIOD (20 Patients)	PRE (50 Patients)	HD (35 Patients)
Total cholesterol (TC) (mmol/L)	4.80 ± 2.64	4.44 ± 1.67	5.12 ± 1.21	4.22 ± 1.30 ^a^
LDL-cholesterol (LDL-C) (mmol/L)	2.40 (1.75–3.03)	2.50 (1.80–4.00)	3.24 (2.28–3.80) ^b,c^	2.45 (1.36–3.00)
HDL-cholesterol (HDL-C) (mmol/L)	1.23 (1.01–1.50)	1.04 (0.95–1.32)	0.88 (0.80–1.81) ^b^	1.06 (0.77–1.21) ^d^
Triacylglycerols (TAG) (mmol/L)	1.63 ± 0.89	1.40 ± 0.53	1.63 ± 0.42	1.44 ± 0.50
Red blood cells (RBC) (10^12^/L)	4.60 ± 0.54	4.70 ± 0.75	3.65 ± 0.60 ^b,c^	3.42 ± 0.50 ^d^^,e^
White blood cells (WBC) (10^9^/L)	8.23 ± 3.44	9.20 ± 2.65	6.72 ± 2.20 ^b,c^	6.54 ± 1.60
eGFR (mL/min/1.73 m^2^)	70.00 ± 18.00	76.90 ± 13.90	25.00 ± 10.70 ^b,c^	7.60 ± 3.14 ^a,d,e^
hsCRP (mg/L)	9.93 (3.41–13.70)	7.54 (3.67–13.56)	8.65 (3.53–11.80)	10.70 (8.60–12.10)

AAA, abdominal aortic aneurysms; AIOD, aortoiliac occlusive disease; PRE, pre-dialyzed group; HD, hemodialyzed group, eGFR, estimated glomerular filtration rate; hsCRP, high-sensitivity C-reactive protein. All data are expressed as mean ± standard deviation or median and interquartile ranges; Significant differences: ^a^ PRE vs. HD; ^b^ AAA vs. PRE; ^c^ AIOD vs. PRE; ^d^ AAA vs. HD; ^e^ AIOD vs. HD; *p* ≤ 0.05.

**Table 3 jcm-09-01416-t003:** The influence of gender on AGEs, RAGEs, AGEs/RAGEs ratio, and UA in the studied groups.

Parameter	AAA			AIOD			PRE			HD		
	Male *n* = 55	Female *n* = 15	*p*	Male *n* = 14	Female *n* = 6	*p*	Male *n* = 27	Female *n* = 23	*p*	Male *n* = 24	Female *n* = 11	*p*
AGEs	17.25 (8.18–38.40)	18.50 (13.45–46.67)	0.2928	29.86 (22.93–56.82)	17.01 (13.65–33.97)	0.0798	1630 (1096–2300)	1665 (1090–1907)	0.8661	2929 (1488–4820)	3027 (1709–5083)	0.5715
sRAGE	54.22 (31.32–127.60)	42.90 (21.43–77.18)	0.2240	96.51 ± 102.60	115.40 ± 76.10	0.6920	3025 ± 1268	2845 ± 800	0.5835	4042 (2887–4139)	3994 (2589–4088)	0.5048
AGEs/sRAGE	0.300 (0.099–0.474)	0.491 (0.229–2.397)	**0.0417**	0.354 (0.276–0.972)	0.264 (0.115–1.581)	0.7441	0.459 (0.322–1.029)	0.606 (0.314–0.794)	0.7359	0.797 ± 0.399	0.969 ± 0.612	0.3549
UA	353.1 ± 121.2	319.50 ± 48.97	0.3146	309.50 ± 98.45	343.60 ± 71.00	0.4925	384.40 ± 107.20	390.40 ± 79.53	0.8351	390.30 ± 92.84	364.10 ± 69.22	0.4489

**Table 4 jcm-09-01416-t004:** The correlation coefficients for AGEs, sRAGE, AGEs/sRAGE, and UA in the studied groups of patients.

**AGEs**				
	***r***	**Group of Patients**	***p* Value**	**95% Confidence Intervals**
sRAGE	0.2801	AAA	0.0217	0.03549 to 0.4931
sRAGE	0.4267	HD	0.0149	0.08090 to 0.6808
UA	−0.4442	HD	0.0394	−0.6339 to −0.01975
sRAGE	0.3271	CKD	0.0031	0.1092 to 0.5150
sRAGE	0.2162	CVD	0.0431	0.000745 to 0.4125
**sRAGE**				
	***r***	**Group of Patients**	***p* Value**	**95% Confidence Intervals**
AGEs	0.2801	AAA	0.0217	0.03549 to 0.4931
AGEs	0.4267	HD	0.0149	0.08090 to 0.6808
eGFR	0.3247	PRE	0.0244	0.03594 to 0.5634
AGEs	0.3271	CKD	0.0031	0.1092 to 0.5150
AGEs	0.2162	CVD	0.0431	0.000745 to 0.4125
eGFR	−0.2302	CVD	0.0386	−0.4325 to −0.00597
	**AGEs/sRAGE**			
	***r***	**Group of Patients**	***p* Value**	**95% Confidence Intervals**
UA	−0.3829	HD	0.0305	−0.6455 to −0.03939
	**eGFR**			
	***r***	**Group of Patients**	***p* Value**	**95% Confidence Intervals**
UA	−0.3210	AAA	0.0056	−0.5183 to −0.09132
UA	−0.3156	HD	0.0392	−0.5627 to −0.01681
sRAGE	0.3247	PRE	0.0244	0.03594 to 0.5634
	**UA**			
	***r***	**Group of Patients**	***p* Value**	**95% Confidence Intervals**
eGFR	−0.3210	AAA	0.0056	−0.5183 to −0.09132
eGFR	−0.3156	HD	0.0392	−0.5627 to −0.01681
AGEs/sRAGE	−0.3829	HD	0.0305	−0.6455 to −0.03939
eGFR	−0.2616	CVD	0.0133	−0.4506 to −0.05016

**Table 5 jcm-09-01416-t005:** Multivariate linear regression analysis of the association between age, gender, hsCRP, and AGEs, sRAGE, UA level in CKD, CVD, and in all the groups studied together.

**CKD**								
**AGEs**			**sRAGE**			**UA**		
**Parameter**	**Coefficient**	***p***	**Parameter**	**Coefficient**	***p***	**Parameter**	**Coefficient**	***p***
age gender hsCRP **eGFR**	−6.386 8.495 16.680 −32.410	0.4523 0.9711 0.1663 **<0.0001**	age gender hsCRP eGFR **AGEs**	–10.810 136.700 2.674 11.030 **0.221**	0.2505 0.5796 0.8401 0.3734 **0.0071**	age gender hsCRP **eGFR** AGEs	−0.311 20.150 −0.749 −1.246 −0.012	0.6348 0.2649 0.4214 **0.0004** 0.0580
**CVD**								
**AGEs**			**sRAGE**			**UA**		
**Parameter**	**Coefficient**	***p***	**Parameter**	**Coefficient**	***p***	**Parameter**	**Coefficient**	***p***
age gender hsCRP eGFR	0.184 −16.210 0.172 −0.009	0.7744 0.2904 0.8211 0.9777	age gender hsCRP **eGFR** **AGEs**	−0.355 80.780 −2.156 −3.062 1.443	0.8592 0.0955 0.3654 **0.0029** **0.0006**	age gender hsCRP **eGFR** AGEs	−0.720 43.080 −0.569 −1.926 −0.066	0.6010 0.1942 0.7272 **0.0061** 0.8089
**ALL**								
**AGEs**			**sRAGE**			**UA**		
**Parameter**	**Coefficient**	***p***	**Parameter**	**Coefficient**	***p***	**Parameter**	**Coefficient**	***p***
age gender hsCRP **eGFR**	–6.386 8.495 16.680 **−32.410**	0.4523 0.9711 0.1663 **<0.0001**	age gender hsCRP **eGFR** **AGEs**	−10.730 −7.769 −12.400 −29.310 0.399	0.1097 0.9663 0.1934 **<0.0001** **<0.0001**	age gender hsCRP **eGFR** AGEs	−0.311 20.150 −0.749 −1.246 −0.012	0.6348 0.2649 0.4214 **0.0004** 0.0580

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
