# Peer review of "Selected Atherosclerosis-Related Diseases May Differentially Affect the Relationship between Plasma Advanced Glycation End Products, Receptor sRAGE, and Uric Acid"

_jcm, 2020, doi:10.3390/jcm9051416_

Round 1
Reviewer 1 Report
The authors addressed all my concerns and questions adequately. The authors are commended on the revised manuscript, which with the added multivariate analyses has gained a lot.
i have a few remaining issues/questions to raise:
I believe a typo has sneaked in on page 11, where authors indicate they adjust for eGFR in the multivariate models, and at the same time test outcome dependence on eGFR. please check and correct.
Multivariate models are not described in the statistics section - please add.
English language has improved, many typos are removed. few remain, e.g. p15 l344 UV should be UA?, p14 l324 in-wall should be in wall?, p16 l 385 "and" is missing, p17 l423 scavenge -> scavenging? p17l426 sentence may need revision-> circulating levels of AGE, sRAGE, UA...? p18 l 434-436 sentence needs a different structure/wording not completely clear.
Author Response
We would like to thank the reviewer for her/his thorough reading of our paper and valuable comments, which have helped us to further improve the quality of the manuscript. Following her/his suggestions, the manuscript has been corrected, and below the changes made are listed point-by-point.
Comments:
The authors addressed all my concerns and questions adequately. The authors are commended on the revised manuscript, which with the added multivariate analyses has gained a lot.
I have a few remaining issues/questions to raise:
- I believe a typo has sneaked in on page 11, where authors indicate they adjust for eGFR in the multivariate models, and at the same time test outcome dependence on eGFR. please check and correct.
The paragraph concerning the multivariate models is corrected.
- Multivariate models are not described in the statistics section - please add.
The sentences listed below are included into the Section “Statistical analysis”:
Additionally, in order to provide greater statistical power and to estimate the independent influence of selected factors on analyzed parameters, multiple linear regression was carried out in the studied groups.
- English language has improved, many typos are removed. few remain, e.g. p15 l344 UV should be UA?, p14 l324 in-wall should be in wall?, p16 l 385 "and" is missing, p17 l423 scavenge -> scavenging? p17l426 sentence may need revision-> circulating levels of AGE, sRAGE, UA...? p18 l 434-436 sentence needs a different structure/wording not completely clear.
The typos listed above have been removed.
The sentence in line 426 has been revised:
In conclusion, the present study has demonstrated that the circulating levels of AGEs, sRAGE, and UA are independently associated with the renal function reflected by eGFR.
The sentence in lines 434-436 has been revised:
The increased UA level in CVD and CKD patients was observed rather due to an impaired kidney function than due to a stimulating effect of high AGEs level.

Reviewer 2 Report
The present study by Bogna Gryszczynska et al. examined the relation between AGE, sRAGE, AGE/sRAGE and UA levels in various disease conditions like abdominal aortic aneurysms, aortoiliac occlusive disease and chronic kidney disease. They report various effects and levels of these components in different disease conditions. The study is well designed but there are some concerns as below.
1) There are studies showing relationship of AGE, sRAGE and UA in CKD and CVD so there is limited scope of novelty.
2) The authors should present the results with subheadings rather than one continuous section.
3) The authors should provide a small separate section mentioning drawbacks of their study.
4) The authors should provide a list for all the abbreviations used.
5) Sentence formation, grammatical and spell check required.
Author Response
We would like to thank the reviewer for her/his thorough reading of our paper and valuable comments, which have helped us to further improve the quality of the manuscript. Following her/his suggestions, the manuscript has been corrected, and below the changes made are listed point-by-point.
Comments:
The present study by Bogna Gryszczynska et al. examined the relation between AGE, sRAGE, AGE/sRAGE and UA levels in various disease conditions like abdominal aortic aneurysms, aortoiliac occlusive disease and chronic kidney disease. They report various effects and levels of these components in different disease conditions. The study is well designed but there are some concerns as below.
- There are studies showing relationship of AGE, sRAGE and UA in CKD and CVD so there is limited scope of novelty.
We agree that the relationship between AGEs and sRAGE in CKD patients is well known. Similarly, the association between hyperuricemia and the progression of kidney disease has been quite frequently discussed. Therefore, we considered the CKD patients to be rather a reference group with known disorders. However, the relation between AGEs, sRAGE, and UA is not completely understood in CKD patients. Nevertheless, concerning CVD, the association mentioned above (or the connection between sRAGE and other ligands) was analyzed in coronary artery disease, heart failure, or acute coronary syndrome [1,2,3]. To the best of our knowledge, there are no studies investigating the effect of the AGE-sRAGE on the disease progression in patients with AAA and AIOD. Generally, oxidative stress is rarely discussed in AAA patients. There are a lot of studies in which the antioxidant enzymes activity, antioxidants level, and level of lipid peroxidation products are measured in serum or tissues of AAA patients [4,5]. The oxidative modification of proteins is rarely undertaken [6]. Another novelty is the correlation between AGEs level, as well as UA level and aneurysm diameter in AAA patients demonstrated in the present study.
Literature:
[1]. Fishman, S. L., Sonmez, H., Basman, C., Singh, V., & Poretsky, L. (2018). The role of advanced glycation end-products in the development of coronary artery disease in patients with and without diabetes mellitus: a review. Molecular Medicine, 24(1), 1-12.
[2]. Ramasamy, R., & Schmidt, A. M. (2012). Receptor for advanced glycation end products (RAGE) and implications for the pathophysiology of heart failure. Current heart failure reports, 9(2), 107-116.
[3]. Fukushima, Y., Daida, H., Morimoto, T., Kasai, T., Miyauchi, K., Yamagishi, S. I., ... & Yamagishi, M. (2013). Relationship between advanced glycation end products and plaque progression in patients with acute coronary syndrome: the JAPAN-ACS sub-study. Cardiovascular diabetology, 12(1), 5.
[4]. Pincemail, J., Defraigne, J. O., Cheramy–Bien, J. P., Dardenne, N., Donneau, A. F., Albert, A., ... & Sakalihasan, N. (2012). On the potential increase of the oxidative stress status in patients with abdominal aortic aneurysm. Redox Report, 17(4), 139-144.
[5]. Miller Jr, F. J., Sharp, W. J., Fang, X., Oberley, L. W., Oberley, T. D., & Weintraub, N. L. (2002). Oxidative stress in human abdominal aortic aneurysms: a potential mediator of aneurysmal remodeling. Arteriosclerosis, thrombosis, and vascular biology, 22(4), 560-565.
[6]. Gryszczyńska, B., Formanowicz, D., Budzyń, M., Wanic-Kossowska, M., Pawliczak, E., Formanowicz, P., ... & Iskra, M. (2017). Advanced oxidation protein products and carbonylated proteins as biomarkers of oxidative stress in selected atherosclerosis-mediated diseases. BioMed research international, 2017.
- The authors should present the results with subheadings rather than one continuous section.
The paragraph “Results” is divided into below-mentioned subsections:
3.1. AGEs, sRAGE, AGEs/sRAGE ratio, and UA level in studied groups.
3.2. The association of AGEs, sRAGE, AGEs/sRAGE ratio, and UA with age, gender, hsCRP, as well as eGFR in studied groups.
3.3. The association of AGEs and UA level with the diameter of the aneurysm in AAA patients.
- The authors should provide a small separate section mentioning drawbacks of their study.
The section “Limitations of the Study” is included into the manuscript.
There are limitations to our study. Firstly, the study was performed in small groups, especially AIOD, and with the unequal number of men and women, which is especially notable in AAA group. This is due to the fact that the male gender constitutes AAA risk factor among the Polish population. Furthermore, the effect of gender on the analyzed parameters was not found in all studied groups. Therefore, despite the small size of the groups and the unequal gender structure, their biochemical and clinical characteristics seem to be well suited to the specificity of these diseases. The hypothesis that an increased AGEs level may stimulate the UA synthesis to help maintain an AGEs-sRAGE balance was not confirmed. It cannot be entirely ruled out that AGEs may stimulate the synthesis of another antioxidant, although additional research is required to verify this concept.
- The authors should provide a list for all the abbreviations used.
The following list of all the abbreviations was added at the end of the manuscript:
Abbreviations:
AAA abdominal aortic aneurysms
AIOD aortoiliac occlusive disease
PRE pre-dialyzed patients (CKD stage 3-4)
HD hemodialyzed patients (CKD stage 5)
CKD chronic kidney disease
CVD cardiovascular disease
AGEs advanced glycation end products
sRAGE soluble receptor for advanced glycation endproducts
UA uric acid
RAGE receptor for advanced glycation endproducts
eGFR estimated glomerular filtration rate
hsCRP high-sensitivity C-reactive protein
ROS reactive oxygen species
RNS reactive nitrogen species
- Sentence formation, grammatical and spell check required.
The whole manuscript has been carefully checked and many language corrections have been made.

Reviewer 3 Report
The authors examined the levels of AGEs, sRAGE, and uric acid in four groups of patients: AAA, AIOD, CKD-PRE, and CKD-HD. They also examined the balance between AGEs and sRAGE by calculating AGEs/sRAGE ratios in these diseases. The hypothesis of the study is that AAA, AIOD and CKD affect the relationship between AGEs, sRAGE, as well as UA levels, due to apparent differences in oxidative stress intensity. They expect to see higher AGEs levels correlate with higher sRAGE levels as well as higher UA levels. Indeed, the authors found the levels of AGEs, sRAGEs are many fold higher in CKD patients than in AAA or AIOD patients. The difference in the levels of UA is mild between different groups, while the ratio AGEs/sRAGE is about two-fold lower in AAA and AIOD group than in CDK-HD group. The nature of the study is very descriptive lacking mechanistic studies. However, this is likely not considered as a major weakness as the Journal of Clinical Medicine would not require mechanistic studies for research in clinical biochemistry?
Comments
- Line 127, plate?
- In CKD-HD patients, only 32.4% patients have NSAID medications. The percentage is much lower than that in other groups. Will the authors consider the potential effects on AGEs, sRAGEs, UA levels?
- Are there differences in AGEs, sRAGEs, UA levels and AGEs/sRAGEs ratio between males and females? Not sure if the authors have addressed this in Table 4.
- The data in Figs. 5 and 6 are interesting. The authors discussed about the negative correlation between UA levels and aneurysm diameter in AAA patients with aneurysm diameter larger than 62 mm. Will the author discuss the data in Fig.5?
Author Response
We would like to thank the reviewer for her/his thorough reading of our paper and valuable comments, which have helped us to further improve the quality of the manuscript. Following her/his suggestions, the manuscript has been corrected, and below the changes made are listed point-by-point.
General comment: The authors examined the levels of AGEs, sRAGE, and uric acid in four groups of patients: AAA, AIOD, CKD-PRE, and CKD-HD. They also examined the balance between AGEs and sRAGE by calculating AGEs/sRAGE ratios in these diseases. The hypothesis of the study is that AAA, AIOD and CKD affect the relationship between AGEs, sRAGE, as well as UA levels, due to apparent differences in oxidative stress intensity. They expect to see higher AGEs levels correlate with higher sRAGE levels as well as higher UA levels. Indeed, the authors found the levels of AGEs, sRAGEs are many fold higher in CKD patients than in AAA or AIOD patients. The difference in the levels of UA is mild between different groups, while the ratio AGEs/sRAGE is about two-fold lower in AAA and AIOD group than in CDK-HD group. The nature of the study is very descriptive lacking mechanistic studies. However, this is likely not considered as a major weakness as the Journal of Clinical Medicine would not require mechanistic studies for research in clinical biochemistry?
Thank you for your comment, but we did not plan the mechanistic studies. The adopted research model did not assume, apart from clinical ones, intervention procedures for the patient’s benefit, for the proper postoperative treatment and good condition following the treatment. The aim of the present study was to evaluate AGEs and sRAGE levels, as well as AGEs/sRAGE ratio in different groups of patients, including those with cardiovascular diseases (AAA and AIOD patients), as well as those with CKD (including PRE and HD patients). We have selected patients with advanced disorders which constituted reference groups for themselves. Therefore, we aimed to assess the influence of increasing oxidative stress accompanying progressive CKD on the studied parameters (PRE patients in comparison to HD patients).
Comments:
- Line 127, plate?
The word “plate” is included in line 127.
- In CKD-HD patients, only 32.4% patients have NSAID medications. The percentage is much lower than that in other groups. Will the authors consider the potential effects on AGEs, sRAGEs, UA levels?
Naturally, the effect of medications listed in Table 1 on the analyzed parameters could not be excluded. However, there is no literature data that would strongly confirm the influence of antihypertensive agents on AGEs and sRAGE levels. Cheng and co-authors [1] estimated the effect of anti-hypertensive drugs of different pharmacological classes, such as Angiotensin Receptor Blockers (ARBs), Angiotensin Converting Enzyme (ACE) inhibitors and Calcium Channel Blockers (CCBs) on AGEs-RAGE axis based on the review of the published articles. Clinical trials delivered mixed results. In most studies, no beneficial treatment effect on AGEs, pentosidine, RAGE, carboxyethyllysine or carboxymethyllysine levels was detected. Even if the reduction of the above-mentioned parameters in blood or urine was observed, various mechanisms proposed to explain this relationship are unlikely. On the other hand, ARB-treated diabetic patients showed AGE-lowering effect and improvements in inflammatory and oxidative stress markers (hsCRP, interleukins-6 and -18) [1,2,3]. The authors postulated that RAGE-associated cellular signaling is inhibited by ARB treatment. To date, the bioactivity of ARBs is unclear. It must be emphasized that in the present study, one of the most critical patient exclusion criteria was diabetes. Some non-steroidal anti-inflammatory drugs (NSAID) increase renal urate excretion, as well as may lead to lover UA serum level. The effect of NSAID on UA level in CKD-HD patients could not be excluded. The lack of significant difference between UA level in HD patients compared to other studied groups may result from drug therapy, as well as from the hemodialysis procedure. There are a few studies on the effect of NSAID on the AGEs level. It was demonstrated that Aspirin has the potential to decrease AGEs accumulation by targeting preformed intermediates, such as pentosidine, by chelation of transition metals, as well as scavenging free carbonyls [4,5]. Additionally, a similar beneficial effect of ACE was found [6]. To the best of our knowledge, there are no studies investigating the impact of NSAID drugs on the sRAGE or AGEs-RAGE axis.
In view of your valuable comment, the sentences below have been included in the section “Limitations of the study”:
Another limitation was the potential effect of anti-hypertensive drugs and non-steroidal anti-inflammatory drugs on the level of the studied parameters. Despite ambiguous data, the impact of medications taken by patients on AGEs, sRAGE and UA levels could not be excluded.
Literature:
[1] Cheng HS, Ton SH, Abdul Kadir K. Therapeutic Agents Targeting at AGE-RAGE Axis for the Treatment of Diabetes and Cardiovascular Disease: A Review of Clinical Evidence. Clin Diabetes Res, 2017, 1(1):16-34.
[2]. Kuboki K, Iso K, Murakami E, et al. (2007) Effects of valsartan on inflammatory and oxidative stress markers in hypertensive, hyperglycemic patients: An open-label, prospective study. Curr Ther Res Clin Exp. 2007, 68: 338-348.
[3].. Persson F, Rossing P, Hovind P, et al. Irbesartan treatment reduces biomarkers of inflammatory activity in patients with type 2 diabetes and microalbuminuria: An IRMA 2 substudy. Diabetes. 2006, 55: 3550-3555.
[4]. Sourris, K. C., Harcourt, B. E., & Forbes, J. M. (2009). A new perspective on therapeutic inhibition of advanced glycation in diabetic microvascular complications: common downstream endpoints achieved through disparate therapeutic approaches?. American journal of nephrology, 30(4), 323-335.
[5]. Urios, P., Grigorova-Borsos, A. M., & Sternberg, M. (2007). Aspirin inhibits the formation of pentosidine, a cross-linking advanced glycation end product, in collagen. Diabetes research and clinical practice, 77(2), 337-340.
[6]. Miyata, T., de Strihou, C. V. Y., Ueda, Y., Ichimori, K., Inagi, R., Onogi, H., ... & Kurokawa, K. (2002). Angiotensin II receptor antagonists and angiotensin-converting enzyme inhibitors lower in vitro the formation of advanced glycation end products: biochemical mechanisms. Journal of the American Society of Nephrology, 13(10), 2478-2487.
- Are there differences in AGEs, sRAGEs, UA levels and AGEs/sRAGEs ratio between males and females? Not sure if the authors have addressed this in Table 4.
The AGEs, sRAGE, UA levels and AGEs/sRAGE in males compared to females have been included in the manuscript.
- The data in Figs. 5 and 6 are interesting. The authors discussed about the negative correlation between UA levels and aneurysm diameter in AAA patients with aneurysm diameter larger than 62 mm. Will the author discuss the data in Fig.5?
Interestingly, the positive correlation between AGEs level and aneurysm diameter lower than 62 mm in AAA patients was found. There are no studies regarding the association between AGEs serum level and the AAA severity, reflected by the aneurysm diameter size. We may hypothesize that oxidative stress, which plays an essential role in the pathogenesis of AAA, is probably more intensified in the early stage of the disease development. This may suggest that oxidative stress has a more significant effect on the development/growth of the aneurysm in the early stages of the disease. In fact, the significance of oxidative stress is possibly decreasing in the advanced stage of the disease. It is possible that an adaptive mechanism against further oxidation may have taken place in AAA patients with a larger aneurysm diameter; however, this theory would require additional experiments. It should be emphasized that AGEs significantly contribute to wall stress. Some authors have suggested that AGEs are involved in arterial stiffening and rigidity in the vessel wall [1,2]. It is difficult to determine what would be the AGEs concentration level in the aneurysm walls. However, it can be assumed that AGEs circulating in the blood do not reflect it, due to the influence of kidneys and the liver role in the exposure to and accumulation of AGEs [3].
Literature:
[1]. Prasad, K.; Mishra, M. Do advanced glycation end products and its receptor play a role in pathophysiology of hypertension? Int. J. Angiol. 2017, 26(01), 001-011.
[2]. Prasad, K. Low levels of serum soluble receptors for advanced glycation end products, biomarkers for disease state: myth or reality. Int. J. Angiol. 2014, 23(01), 011-016.
[3]. de Vos, L. C., Lefrandt, J. D., Dullaart, R. P., Zeebregts, C. J., & Smit, A. J. (2016). Advanced glycation end products: An emerging biomarker for adverse outcome in patients with peripheral artery disease. Atherosclerosis, 254, 291-299.

This manuscript is a resubmission of an earlier submission. The following is a list of the peer review reports and author responses from that submission.
Round 1
Reviewer 1 Report
This paper is based on a detailed biochemical study, sometime difficult to understand for physicians, also because of the big numbar of abbreviations.
The main question addressed concerns a marker of atherosclerosis in a particular aortic vascular pathology.
The topic is original, although with a particular interest in the biochemistry field. Under this aspect, original contributions can be found.
The text is long , very detailed in biochemical particulars , and with a lot of abbreviations: I suggest to change it in a simple version for common physicians.
The conclusions are appropriate; however their practical impact could be specified. In fact they address to the main biochemical questions posed.
Could you improve the facility of reading for person , not particularly skilled in this chapter of biochemistry?
Reviewer 2 Report
Review of
Aortic abdominal aneurysms and atherosclerosis related diseases affect the relationship between advanced glycation end products, receptor sRAGE
By Gryszczynska et al,
In general: language revision is strongly recommended. Please read carefully every sentence and check for sentence structure and message of the exact sentence. E.g. Line 68-70: sentence does not make sense.
– please ask for assistance by a native English-speaking person.
That being said, I find the perspectives of the study very interesting, although revision is required before I can recommend the paper for publication. Mainly due to unclear decisions about statistical analyses and unclear qualification of subgrouping and analyses of data within subgroups. The study material and findings deserve attention. Given the size of the dataset, the authors are recommended to find support by a biostatistician to gain full value of their precious dataset.
Abstract
Hypothesis is missing, the level of uric acid is included as outcome, but there is no qualification of why.
Data on UA are not mentioned, but relation of UA to AGE levels is elaborated upon in the conclusion of the abstract. Needs revision!
Introduction
References are needed to support the statement lines 44-46!
In general: language revision is strongly recommended. Please read carefully every sentence and check for sentence structure and message of the exact sentence. E.g. Line 68-70: sentence does not make sense.
Should line 71 have been deleted?
line 89-90: why is it interesting to study AGE/RAGE in AAA and AIOD?
Hypothesis is missing
Aim clear
I miss qualification of why the 4 different patient groups were chosen for the study. Atherosclerosis driven AAA? Atherosclerosis may facilitate and may counteract the development of AAA, depending on the exact pathology of the individual patient,
Did results differ between pt subgroups with more atherosclerosis and less atherosclerosis?
how were arteries from the patient groups evaluated for atherosclerosis?
When outcome was adjusted for renal function and/or low grade inflammation, did groups then differ significantly (multivariate statistical analyses)? What about when analyses are adjusted for biological sex and age too?
If 20 mg/dL promotes EC dysfunction (UA induces endothelial dysfunction by activating the HMGB1/RAGE signalling pathway, Biomed Res Intl, Cai et al 2017) - how does the measured plasma UA levels correspond hereto? May the elevated UA be a cause of the disease, rather than a protective factor?
Statistical analyses: three tests for normality are listed, Pearson omnibus was given the highest priority acc the producers’ recommendation. Was any concern/thought given the sensitivity of each of the tests? And the nature of the data (patient data)? In my experience, including a statistician in the research group is a great help on these parts, also acc the question re multivariate analyses above. For this type of data, Shapiro Wilks will possibly be the best option.
Was an attempt done to e.g. Transform non-normal data to gain normal distribution?
Please indicate exactly per analysis which method was used. In statistics M&M section or in figure legends.
Results
Are the higher levels of AGE, sRAGE in the CKD patients due to the decreased AGE secretion by the kidneys? See again my question on multivariate analyses -possibly it will also be of added value to subgroup AAA and AIOD in one group, and the CKD patients in another for the multivariate analyses. From figure 1, quite some variability in the groups exist (as expected with patient data). Studying the AAA vs AIOD groups will also be relevant cf your discussion lines 291-296.
Please provide exact p values.figs 1+2+3
Please present UA levels in a similar way as AGE and sRAGE levels (i.e. In a figure) to show the distribution of the data obtained.
UA kind of pops out of the blue in the corr coeff table.
Was the sensitivity of the Pearson or SPearman correlation towards non-normal distributed data considered before running the correlation analyses?
When already having performed the correlation analyses, why was the data then next treated as categorical variables? All seems to me to be continous variables, and the correlation analyses will be enough?! It would be great when the authors could show the graphs with the correlations + confidence intervals. And in the table 3 provide the exact p-values.
It would good sense to divide the patients acc renal failure group 1-5, and then perform comparative analyses on the outcomes, rather than divide the patients per outcome quartile. And then for the correlation analyses perform the multivariate analyses, cf. Above.
Line 226, I do not understand the “the classification of AAA patients into appropriate subgroups….. did not differ significantly”? What did not differ? Please clarify.
The finding of AAA diameter >62 mm correlating negatively with UA concentration is highly interesting. Did you compare patients’ charateristics for the two AAA subgroups? Were the patients different in any respects?
Were any attempts done to obtain AAA tissue from surgeries in order to quantify tissue levels of ROS (cf ref 25), AGEs and UA?
Lines 245-247 AGEs are on earlier occassions shown to be increased in CKD patients due to the lower renal filtration. Please consider revising this statement, also cf your discussion lines 253-264.
Does your conclusion line 273-274 change with re-analysis cf above comments (normal distribution or multivariate test)?
Please elaborate on the AGE and sRAGE vs UA analyses in all 4 different pt groups, in light of UA possibly promoting the disease. Cf multivariate analyses also.
Please discuss local tissue presence vs plasma measurements, i.e. The value of a blood sample to predict local tissue metabolism (if possible at all).
Conclusion
Lines 398-399. Please revise sentence. You do not provide data to support a mechanistic / cause consequence explanation, but correlation analyses. In nature, your study is cross sectional, i.e. Be careful when writing the conclusion.
lines 399-400: All diseases are atherosclerosis related: how did you measure the degree of atherosclerosis (cf question above)? Or did you assume this, on basis of literature?
Are any studies available where the effect of antioxidants on disease status/progression in any of your 4 patient groups is investigated? Would be great to add in a perspectives section.